# Biological Mechanisms Involved in Muscle Dysfunction in COPD: An Integrative Damage–Regeneration–Remodeling Framework

**DOI:** 10.3390/cells14211731

**Published:** 2025-11-04

**Authors:** Joaquim Gea, Mauricio Orozco-Levi, Sergi Pascual-Guàrdia, Carme Casadevall, César Jessé Enríquez-Rodríguez, Ramon Camps-Ubach, Esther Barreiro

**Affiliations:** 1Respiratory Medicine Department, Hospital del Mar Research Institute, Área de Enfermedades Respiratorias, CIBER, ISCIII, 08003 Barcelona, Spain; mauricio.orozco.levi@gmail.com (M.O.-L.); spascual@hmar.cat (S.P.-G.); cesarjesse.enriquez01@alumni.upf.edu (C.J.E.-R.); ramon.camps.ubach@hmar.cat (R.C.-U.); ebarreiro@researchmar.net (E.B.); 2MELIS Department, Universitat Pompeu Fabra, 08003 Barcelona, Spain; 3Respiratory Medicine Department, Fundación Cardiovascular de Colombia (FCV), Hospital Internacional de Colombia (HIC), Universidad de Santander (UDES), Floridablanca 681004, Santander, Colombia

**Keywords:** muscle dysfunction, inflammation, oxidative stress, injury, deconditioning, hyperinflation

## Abstract

Skeletal muscle dysfunction is a major systemic manifestation of COPD that shapes symptoms, exercise tolerance and mortality. Current evidence can be integrated within a Damage–Regeneration–Remodeling framework linking mechanics and biology to clinical phenotypes. Pulmonary hyperinflation and chest wall geometry chronically load the diaphragm and other respiratory muscles in COPD, whereas inactivity and exacerbation-related disuse underload locomotor muscles. Across muscle compartments, oxidative/nitrosative stress, activation of proteolytic pathways, mitochondrial and endoplasmic reticulum stress, microvascular limitations, neuromuscular junction instability, and myosteatosis degrade muscle quality. The diaphragm adapts with a fast-to-slow fiber shift, greater oxidative capacity, and sarcomere foreshortening, improving endurance, whereas limb muscles show atrophy, a glycolytic shift, reduced oxidative enzymes, extracellular matrix accrual, and fat infiltration. Translational levers that address these mechanisms include: (I) *Reduce damage*: bronchodilation, lung-volume reduction, oxygen, non-invasive ventilation, early mobilization, pulmonary rehabilitation, neuromuscular stimulation, and corticosteroid stewardship; (II) *Enable regeneration*: progressive resistance plus high-intensity/heavy-load endurance training; adequate protein and vitamin-D intake, and endocrine correction; and (III) *Steer remodeling*: increase physical activity (with/without coaching/telecoaching), functional assessment and CT or MRI monitoring, inspiratory-muscle training, and phenotype-guided adjuncts in selected cases. This framework clarifies why lung deflation strategies benefit inspiratory mechanics, whereas limb recovery requires behavioral and metabolic interventions layered onto systemic optimization.

## 1. Introduction

Chronic obstructive pulmonary disease (COPD) is a highly prevalent disorder, which results from long-term exposure to harmful inhaled noxious agents, such as tobacco smoke or ambient particulates [1]. Clinically, COPD is characterized by respiratory symptoms and chronic, poorly reversible, often progressive airflow obstruction [1] as well as heterogeneous clinical presentations, denominated phenotypes. Some of these phenotypes can be linked to specific clinical management strategies and even to targeted treatments; accordingly, they have been termed ‘treatable traits’ [2]. COPD is associated with an abnormal pulmonary and systemic inflammatory response, which partly explains why clinical manifestations are not confined to the lungs but extend to other organs and tissues [3,4]. One of the most frequent and clinically relevant systemic manifestations of COPD is skeletal muscle dysfunction, which emerges from the convergence of mechanical, metabolic, and inflammatory stressors that injure myofibers and their support systems, constrain repair/regeneration, and steer remodeling toward adaptive or maladaptive trajectories. Although respiratory muscles (with evidence primarily from the diaphragm) and limb muscles (typically the quadriceps) share multiple molecular signatures (oxidative/nitrosative stress, proteostasis imbalance, and mitochondrial dysfunction) [5,6], they differ in their mechanical loading environments and phenotypic outcomes. Indeed, the chronically loaded diaphragm displays oxidative adaptations under hyperinflation, whereas the relatively disused lower limb muscles tend to undergo atrophy, lose oxidative capacity, and accrue myosteatosis [5,6,7]. This narrative review focuses on synthesizing the cyclic Damage–Regeneration–Remodeling mechanisms that most directly explain clinical phenotypes in COPD (Table 1).

## 2. Damage (Injury and Molecular Derailment)

### 2.1. Mechanical Load-Related Injury

Static and dynamic pulmonary hyperinflation (the former is derived from air trapping at rest, whereas the latter adds a volume increase during exertion) shortens diaphragm fibers, flattens the dome, and reduces the zone of apposition. This shifts the muscle away from its optimal length–tension relationship and, consequently, reduces pressure generation for a given neural drive [8,9,10]. The operating length of intercostal/parasternal muscles is also altered by changes in rib cage configuration and intrinsic positive end-expiratory pressure (PEEPi) [8,11,12]. These geometric constraints, superimposed on airway obstruction, act as chronic threshold loads on inspiratory muscles in COPD [8,11]. In contrast, lower-limb disuse, accentuated around exacerbations, drives limb muscle deconditioning [6,13,14]. Together, these factors explain depressed inspiratory and lower-limb muscle strength for a given ventilatory drive, while shared biological mechanisms (oxidative stress, proteolysis, bioenergetic impairment; see in the following sections) operate in both compartments [5,15,16,17].

### 2.2. Oxidative and Nitrosative Injury

Excess reactive oxygen/nitrogen species (ROS/RNS) derived from systemic inflammation, hypoxemia/reperfusion phenomena, and local mitochondrial NADPH oxidases (NOX) sources induce DNA, protein, and carbohydrate modifications, which disrupt excitation–contraction coupling (ryanodine receptor [RyR1]/SERCA), Ca^2+^ handling, and cross-bridge kinetics in respiratory and limb muscles [15,18,19,20,21]. In severe COPD, oxidative signatures are consistently elevated in respiratory muscle and variably in limb muscle, where heterogeneity likely reflects disease stage and activity [15,18,22,23,24,25].

### 2.3. Proteostasis Failure-I: Mismatching Between Protein Synthesis and Proteolysis—Ubiquitin–Proteasome System-, Autophagy-, and Ca^2+^-Dependent Pathways

In skeletal muscle, loss of proteostasis mainly results from the coordinated activation of proteolytic pathways. The ubiquitin–proteasome system (UPS) is upregulated via the E3 ligases atrogin-1 (MAFbx) and Muscle RING finger-1 (MuRF-1), which ubiquitinate myofibrillar and regulatory proteins for degradation by the 26S proteasome [26,27]. In parallel, the autophagy–lysosome pathway is modulated in a context-dependent manner: increased LC3-II together with reduced p62 indicates enhanced autophagic flux [28]. Ca^2+^-dependent calpains, along with caspases, cleave sarcomeric proteins and generate fragments that are subsequently removed by the UPS and/or autophagy, thereby amplifying proteolysis and contributing to muscle dysfunction [29,30,31].

In the quadriceps of patients with COPD, higher *atrogin-1/MuRF-1* transcripts and FoxO1 signaling are associated with atrophic phenotypes [26,32]; however, this upregulation is not universal at the protein level, underscoring heterogeneity by phenotype, disease stage, hypoxemia, disuse/inactivity, nutrition, and corticosteroid exposure [6,33,34,35]. Acute exacerbations transiently intensify these catabolic programs, and repeated episodes can drive cumulative, stepwise losses in muscle mass and function [6,34,36].

### 2.4. Mitochondrial and Quality Control

COPD lower-limb muscles exhibit reduced mitochondrial content and electron-transport/respiratory-chain activity, with decrements in oxidative enzymes such as citrate synthase (CS), succinate dehydrogenase (SDH), and cytochrome c oxidase (COX), mitochondrial DNA (mtDNA) damage, and impaired organelle quality control [15,37,38,39,40,41,42]. Imbalances in mitochondrial fusion–fission dynamics and mitophagy, together with early downregulation of oxidative programs (e.g., PGC-1α/NRF-1/TFAM), can precede or accompany the shift from oxidative toward more glycolytic fibers [43,44]. Once this shift becomes established, it will further depress mitochondrial density and respiratory capacity, creating a self-reinforcing loop. Functionally, the resulting limitation in fatty-acid β-oxidation would promote intramyocellular lipid deposition and accumulation of lipotoxic intermediates (e.g., diacylglycerols, ceramides), thereby driving myosteatosis [45]. This, in turn, will result in further energetic capacity, and impaired excitation–contraction coupling and insulin-sensitive metabolism [46,47]. By contrast, changes in upper-limb muscles are generally less pronounced [48,49]. The chronically loaded diaphragm can display partial mitochondrial enrichment and oxidative adaptations [50], yet its function remains quality-limited by coexisting systemic and local oxidative stressors [51,52,53].

### 2.5. Proteostasis Failure-II: Endoplasmic Reticulum Stress and Unfolded Protein Response

Markers and pathways of endoplasmic-reticulum (ER) stress and unfolded protein response (UPR) are implicated in COPD skeletal muscle dysfunction, mechanistically linking redox burden with proteostasis failure. Persistent oxidative stress and low-grade inflammation can disrupt ER homeostasis and trigger UPR activation in limb muscles, initially as an adaptive attempt to restore protein folding and translational balance. However, when sustained, this response may become maladaptive, suppressing anabolic signaling through inhibition of mTOR and Akt pathways, promoting proteasome- and autophagy-mediated protein degradation, and thereby contributing to muscle atrophy. Consistent with this upstream role of redox burden, preclinical work shows that NOX-derived ROS following cigarette smoke exposure directly disrupt proteostatic signaling in limb muscle; conversely, NOX inhibition with apocynin preserved IGF-1/mTOR signaling and mitigated muscle atrophy, underscoring oxidative stress as a tractable driver of proteostasis failure [54]. Evidence from quadriceps samples supports such activation, in parallel with enhanced autophagy–lysosome flux and altered chaperone expression [40,41,55,56]. By contrast, data for the chronically loaded diaphragm are less consistent, with one study reporting no UPR induction in stable COPD [56]. This lack of activation could relate to cohort characteristics (namely, patients with relatively preserved body composition), which may have limited the degree of ER stress experienced by this muscle. Nevertheless, further studies are required to confirm whether UPR signaling in the diaphragm differs qualitatively or quantitatively from that observed in limb muscles.

### 2.6. Microvascular and Neuromuscular Junction Contributors

Patients with COPD show lower capillary-to-fiber ratios and microvascular/endothelial dysfunction in locomotor muscles, limiting O_2_ delivery (DO_2_), particularly during exercise [57,58,59,60]. Training-induced angiogenic remodeling is often blunted, which may cap performance gains [61]. Moreover, quadriceps biopsies frequently display fiber-type grouping and small angular fibers, consistent with denervation-reinnervation dynamics; experimental data also show neuromuscular junction (NMJ) instability and degeneration (e.g., acetylcholine receptor [AChR] fragmentation, denervation-related transcript signatures) [16,62,63], a phenomenon that can also be triggered by cigarette smoking [62]. Mechanistically, NMJ instability reduces the safety factor for neuromuscular transmission; during repeated contractions this leads to activity-dependent transmission failure, asynchronous motor-unit recruitment, and larger reinnervated units that require higher central drive, thereby increasing fatigability beyond what would be expected from atrophy alone. This may also dampen rehabilitation responsiveness, with emerging NMJ-related biomarkers (e.g., circulating C-terminal agrin fragment [CAF22], brain-derived neurotrophic factor [BDNF], glial cell line-derived neurotrophic factor [GDNF]) linking synaptic stability to functional recovery during pulmonary rehabilitation [16,62,63,64].

From a hemodynamic standpoint, exercise muscle blood flow reflects the product of perfusion pressure (≈mean arterial pressure, MAP) and microvascular conductance. Thus, interventions that improve endothelial function yet lower systemic blood pressure may yield a neutral or even negative net effect on exercising-muscle perfusion. In the randomized trial by Curtis et al., ACE-inhibitor therapy reduced systolic blood pressure but attenuated the peak work-rate improvement with training compared with placebo, aligning with the idea that in COPD, given the microvascular rarefaction and blunted angiogenic plasticity, the drop in perfusion pressure can outweigh potential conductance benefits, thereby constraining performance gains [65]. As mentioned in the previous section, overall muscle abnormalities in upper-limb muscles are generally less pronounced than in the legs [23,66]. The chronically loaded diaphragm, in turn, often shows capillary enrichment which, together with improvements in mitochondrial oxidative capacity, indicates aerobic adaptation [51,67,68]. However, its efficacy remains quality-limited by coexisting systemic stressors (e.g., oxidative endoplasmic reticulum stress, hyperinflation-related strain) [51,68,69]. Evidence from our group also shows increased capillarity not only in the diaphragm but also in the external intercostals of COPD, supporting remodeling of respiratory-muscle microvasculature [70].

### 2.7. Lipotoxicity and Myosteatosis

Intramuscular fat and toxic lipid intermediates (e.g., ceramides, diacylglycerols) impair mitochondrial function and disturb excitation–contraction coupling, degrading “muscle quality” even when mass is relatively preserved [71]. Computed-tomography muscle attenuation (Hounsfield units) and magnetic resonance imaging (MRI) proton-density fat fraction (PDFF) quantify myosteatosis [72,73,74]. Higher fat content (e.g., lower CT attenuation, higher PDFF) relates to worse strength, walking capacity and bioenergetics in COPD limb muscles [75,76]. Moreover, lower muscle density or CT-derived muscle indices and fat-free mass indices predict mortality in several cohorts, emphasizing that quality can matter as much as quantity [77,78,79].

Notably, increased intramuscular lipid content is not always pathological. It can also be found in endurance-trained athletes, where lipid droplets are mainly stored in oxidative fibers and efficiently mobilized during exercise [80]. In contrast, in COPD muscles, lipid accumulation reflects impaired mitochondrial oxidation and dysregulated lipid turnover, leading to the buildup of bioactive and potentially toxic lipid species. Differences in droplet localization, composition, and metabolic regulation likely explain these opposing functional outcomes.

## 3. Regeneration and Their Bottlenecks

### 3.1. Satellite Cells and Myogenesis

Sarcopenic COPD shows smaller Pax7^+^ satellite cell pools and defects in activation and differentiation (MyoD/myogenin), limiting myofiber replacement after injury and even after training stimuli [81,82]. Similar results have been observed in animal models of emphysema [83]. These myogenic deficits are aggravated by the COPD systemic milieu (oxidative/inflammatory stress, hypoxemia, corticosteroid exposure, and comorbidities), which blunts myogenic transcription programs (e.g., TNF-α/myostatin effects on MyoD/myogenin) and can directly impair muscle precursor cell differentiation, particularly during acute exacerbations [15,16,84,85,86,87]. Clinically, this helps explain the incomplete recovery of muscle mass and strength after exacerbations or immobilization associated with bed rest or hospitalization. Indeed, quadriceps force falls during admission and often recovers only partially over weeks to months, despite rehabilitation [88,89,90,91].

### 3.2. Inflammatory Orchestration of Muscle Regeneration

Effective muscle regeneration depends on orderly inflammatory resolution and timely macrophage polarization from the pro-inflammatory phenotype (M1) to the pro-regenerative one (M2) [15,16]. In COPD, persistent systemic inflammation and recurrent acute exacerbations disrupt these processes and steer regeneration toward fibrotic remodeling with incomplete recovery of muscle function [85,88,90]. Moreover, several intramuscular cytokines (myokines) seem to behave as context-dependent, double-edged signals: when transient and proportionate, they would activate myogenic programs, supporting regeneration and adaptive remodeling. By contrast, sustained or excessive cytokine expression would drive contractile dysfunction, proteolysis, and maladaptive remodeling [92]. In COPD this behavior pattern is muscle-specific: external intercostals show upregulation of proinflammatory cytokines, such as tumor necrosis factor-α (TNF-α) and interleukin-6 (IL-6), and TNF-receptor expression relates to paired box protein 7 (Pax7), M-cadherin, and MyoD, consistent with a pro-regeneration signaling axis [93,94]. In contrast, TNF-α and its receptor TNFR2 are reduced, and the levels of the former correlate directly with muscle strength [95].

### 3.3. Growth Signaling and Anabolic Resistance

In COPD muscle, impaired anabolic signaling (reduced insulin-like growth factor-1 [IGF-1]/Akt [proteinkinase B]/mTOR [mechanistic target of rapamycin] activity together with overactive transforming growth factor-β [TGF-β] family brakes [myostatin/activin]) suppresses protein synthesis and myogenesis [15,16,85]. The magnitude of this anabolic response is further shaped by age, sex, and the endocrine milieu [16,85]. Correcting reversible endocrine-nutritional deficits (e.g., hypogonadism and vitamin D deficiency) and ensuring sufficient protein intake (including leucine) alongside progressive resistance training can partially re-sensitize these pathways and support myofiber accretion [16,85,91]. Consistently, pulmonary rehabilitation increases quadriceps IGF-I and myogenic regulatory factor (MyoD) expression in COPD, and (at least in non-cachectic patients) downregulates myostatin, thereby reactivating anabolic/myogenic programs [96,97].

## 4. Remodeling (Adaptive vs. Maladaptive Trajectories)

### 4.1. Respiratory Muscles: Predominant Adaptive Oxidative Remodeling Under Chronic Load

The human COPD diaphragm shows a fast-to-slow fiber shift together with enhanced oxidative capacity (increased mitochondrial content and, in many series, greater capillary supply); sarcomere length is reduced, consistent with chronic foreshortening [23,50,51,98,99]. These endurance-oriented adaptations improve fatigue resistance but do not restore maximal pressure generation because hyperinflation-related geometry (shortened zone of apposition and flattened curvature) persists [8,51]. Clinically, this helps to explain the dissociation between relatively preserved or even increased inspiratory endurance and depressed peak pressure generation [51,100]. However, when tested at matched lung volumes (i.e., in the hyperinflated range), patients with COPD can generate higher transdiaphragmatic pressure than healthy subjects [10], corroborating their adaptation. With respect to other respiratory muscles, there is evidence of structural damage in the external intercostals, coexisting with activation of myogenic pathways and satellite cells in patients with COPD. This is associated with increased expression of certain myokines, likely implicated in both processes [93,94,101]. Interfibrillar capillary density also appears to increase [70], while the enzymatic capacity of both oxidative and glycolytic pathways is roughly maintained or even enhanced [102,103,104]. Overall, the net effect seems to be maintenance or a slight increase in the proportion of type II fibers and myosin heavy-chain isoform IIa [105,106]. In contrast, changes in parasternal muscles appear to be similar to those observed in the diaphragm, with a shift toward an increased proportion of slow-twitch type I fibers [107]. Why might intercostals retain more effective myogenic signaling despite proximity to pulmonary inflammation? Several non-mutually exclusive factors likely contribute: (i) a continuous loading pattern and higher neural drive that sustain anabolic/mechanotransductive cues; (ii) relatively preserved innervation and duty-cycle activity, limiting disuse atrophy; (iii) greater capillary density and oxidative support, which favor satellite-cell function; and (iv) a compartmentalized microenvironment (pleural and fascial interfaces) that may buffer spillover from parenchymal inflammation.

### 4.2. Limb Muscles: Glycolytic Shift and “Quality Loss”

By contrast, the quadriceps and other locomotor muscles commonly show fiber atrophy with a shift toward faster type II phenotypes, together with reduced mitochondrial content and oxidative enzyme capacity and lower capillary density [6,39,108]. Myosteatosis is also frequent and contributes to impaired bioenergetics and worse muscle function by disrupting mitochondrial oxidative metabolism, promoting accumulation of lipotoxic intermediates, and interfering with excitation–contraction coupling and insulin signaling [101,102,103]. Functionally, this maladaptive remodeling magnifies fatigability, slows muscle O_2_ utilization (VO_2_) kinetics, and limits endurance [45,46,109,110]; these effects are compounded when physical inactivity persists after acute exacerbations [13,90].

### 4.3. Extracellular Matrix and Fibrosis. Neuromuscular Junction Remodeling

In COPD limb muscles, chronic low-grade myofiber injury, characterized by sarcomeric and sarcolemmal disruptions, segmental necrosis, and localized inflammatory infiltration, has been documented by ultrastructural analyses [111]. These lesions likely result from repetitive mechanical strain during coughing and breathing against increased airway resistance, compounded by oxidative and proteolytic stress and the vulnerability associated with inactivity. Over time, such repeated micro-injury–regeneration cycles promote intramuscular extracellular matrix (ECM) deposition and stiffening, disrupting lateral force transmission along the fiber–ECM–tendon chain and slowing functional recovery [112,113]. In the quadriceps, transcriptomic and histologic studies confirm ECM-program activation and increased intrafiber ECM proteins in COPD, consistent with fibrosis-driven mechanical impairment [114,115]. Notably, transcriptomic meta-analyses comparing disuse atrophy and resistance exercise-induced hypertrophy indicate that COPD-related muscle dysfunction is not merely the inverse of training adaptation, but rather involves distinct molecular disruptions that may require targeted therapeutic strategies beyond conventional rehabilitation [116]. Furthermore, NMJ degeneration reduces transmission safety and slows rate of force development, degrading gait/balance and increasing fall and injury risk in COPD.

At the neuromuscular junction, remodeling can stabilize motor units (via reinnervation) or fail (resulting in persistent denervation). Smoking-induced neuromuscular junction degeneration and biomarker signatures in COPD support a contribution to variable motor-unit recruitment and day-to-day performance [62,64]. Quadriceps stiffness measured by shear-wave elastography correlates with functional indices, linking ECM remodeling to performance variability [117].

### 4.4. Time Course and Partial Reversibility

Longitudinal observations indicate stepwise decrements around exacerbations [36,118,119], with partial reversibility under targeted rehabilitation, oxygenation/deflation, and activity coaching [88,120]; biologically targeted measures (redox, proteostasis, endocrine/nutrition) are often required to consolidate gains [121].

## 5. Translational Implications Mapped to Damage–Regeneration–Remodeling

### 5.1. Reduce Damage in Respiratory Muscles

Within a Damage-Regeneration-Remodeling framework, the first lever is to reduce damage. Dual long-acting bronchodilators (long-acting beta-agonists, long-acting anticholinergics) reduce static and dynamic hyperinflation, and improve inspiratory mechanics and exercise tolerance [1,122]. In eligible emphysema/hyperinflation phenotypes, lung-volume reduction (surgical [LVRS] or through bronchoscopy [endobronchial valves]) further decreases hyperinflation and improves function [123,124]. Moreover, home non-invasive ventilation (NIV) in persistent hypercapnic COPD can mitigate gas exchange stress and the mechanical burden of dynamic intrinsic PEEPi during exertion and/or sleep [11,125,126].

### 5.2. Reduce Damage in Limb Muscles

In stable COPD patients, individualized exercise training (aerobic and resistance), is the cornerstone to preserve peripheral muscle and improve strength and endurance [121,127,128,129]. In appropriate candidates, supplemental oxygen during training can reduce exercise-induced hypoxemia and related redox/HIF-1α-mediated inhibitory signaling (which contributes to myofiber injury and catabolism), improve O_2_ delivery to locomotor muscles, and, most importantly, enable higher training workloads that drive protective oxidative and hypertrophic adaptations [130,131,132]. Moreover, non-invasive ventilatory support during training sessions lowers the work of breathing and sympathetic vasoconstriction, redistributing blood flow toward locomotor muscles and thereby blunting post-exercise quadriceps fatigue in severe COPD [133]. During (still debated) and after exacerbations, early mobilization and early pulmonary rehabilitation limit disuse atrophy and accelerate functional recovery [134,135]. Finally, in very deconditioned patients or during and following exacerbations, neuromuscular electrical or magnetic stimulation can help preserve or regain strength when active exercise is limited [136,137,138,139,140].

In addition, other measures such as correcting gas exchange abnormalities, endocrine-nutritional care, and steroid stewardship can modify factors affecting both respiratory and limb muscles. In this regard, long-term home oxygen therapy (LTOT) improves survival in patients with severe resting hypoxemia [125]; adding home non-invasive ventilation to oxygen prolongs time to readmission or death in persistent hypercapnia [126]. Both approaches reduce the mechanical burden of dynamic intrinsic PEEPi [11]. In male patients with documented hypogonadism, testosterone plus resistance training can increase lean mass and strength [141,142]. Similarly, correcting vitamin D deficiency and ensuring adequate protein/energy intake can improve muscle function in selected cases [143,144]. Leucine/branched chain amino acid (BCAA) supplementation has not consistently outperformed complete protein intake [145,146]. Avoiding prolonged systemic courses outside acute exacerbations and using the lowest effective dose with limited duration to reduce the risk of glucocorticoid-induced myopathy [1,147,148].

### 5.3. Enable Regeneration

Progressive resistance training combined with high-intensity interval or heavy-load endurance work is recommended to support myogenesis, hypertrophy, and to stimulate mitochondrial biogenesis, greater capillary density and improved oxidative capacity. Together, these modalities are associated with myofibrillar accretion, enhanced myogenic signaling, and restoration of aerobic metabolism in the quadriceps of COPD patients [91,129,149,150]. An overall protein intake of around 1.2–1.5 g·kg^−1^ per day in adults is advised, with emphasis on high-quality, leucine-rich sources distributed across meals [151,152]. However, adding isolated leucine to a protein dose has not consistently outperformed complete protein supplementation in COPD [145]. Correction of vitamin D deficiency and management of endocrine deficits (e.g., hypogonadism) may further augment gains in lean mass and strength when combined with training [141,142,143].

### 5.4. Steer Remodeling

Sustaining pulmonary-rehabilitation gains is favored by integrating structured physical-activity coaching/telecoaching into follow-up to reduce re-disuse; step counts improve, although maintaining fitness benefits typically also requires attention to exercise intensity and progression [153,154]. Monitoring muscle mass and myosteatosis is feasible with CT muscle radiodensity and MRI fat-fraction; higher intramuscular fat relates to worse function and bioenergetics, and these metrics can help tailor exercise/nutrition strategies [45,46,77,118]. For inspiratory muscle weakness or high ventilatory loads, adding inspiratory-muscle training in the pulmonary rehabilitation framework reduces dyspnea and improves maximal inspiratory pressures and health status. Indeed, inspiratory muscle training induces structural adaptations in the inspiratory muscles themselves, accompanied by functional improvements in both strength and endurance [155]. Typical effective protocols commonly use sessions of around 20 min, more than 3 times per week, at 30–60% of maximal inspiratory pressure [156,157,158]. Other potential adjuncts (e.g., antioxidants or anabolic drugs, such as androgenic substances or growth hormone secretagogues) remain investigational and are best limited to phenotype-guided use within trials or specialist oversight while evidence matures [6,15].

## 6. Summarizing: What Differs Between Respiratory and Limb Muscles, and Why Does It Matter? (Figure 1)

Both ‘compartments’ (respiratory muscles and limb muscles) share core mechanisms of muscle dysfunction in COPD: oxidative stress, systemic inflammation, activation of proteolytic pathways, impaired bioenergetics, biological effects of drugs, gas exchange abnormalities and nutritional deficiencies, etc. [5,6,7,15,16,85,159]. Even so, the diaphragm (chronically loaded yet mechanically disadvantaged by hyperinflation) develops a slower-twitch and more oxidative fiber phenotype, resulting in a fatigue-resistant functional profile at the cost of decreased maximal force [8,10,50,51,98]. By contrast, locomotor muscles are underloaded in daily life due to reduced physical activity, and drift toward a glycolytic profile (type II fiber predominance), lower-quality muscle with reduced oxidative capacity and increased intramuscular fat [6,45,46,108]. This divergence helps explain why lung deflation and ventilatory strategies can acutely improve chest-wall and lung mechanics and respiratory muscle function [1,11,123,124], whereas reversing limb muscle dysfunction generally requires behavioral (training, more physical activity) and metabolic (nutrition, anabolic stimuli) levers layered onto systemic optimization [121,127].

**Figure 1 cells-14-01731-f001:**
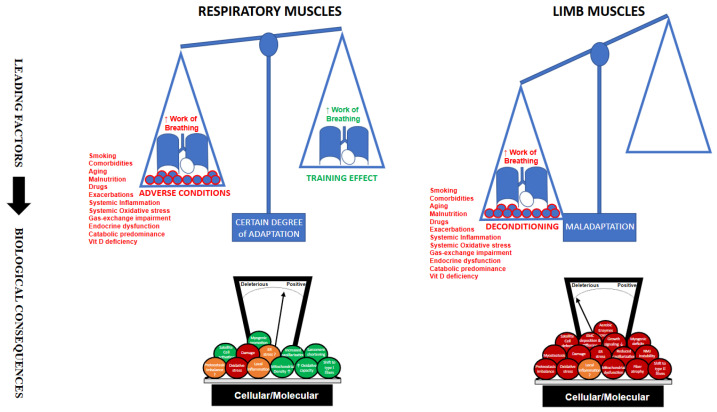
The figure shows the most influential determinants acting on the respiratory and limb muscles of patients with COPD, as well as their biological consequences, which will ultimately shape muscle structure and function. Whereas the features of the respiratory muscles result from both positive and negative influences—though with a predominance of the latter—only negative factors seem to act on the limb muscles (particularly on the lower limbs), which accounts for their poorer phenotype. The arrow indicates that the leading factors determine the biological consequences.

## 7. Conclusions

COPD muscle dysfunction arises from converging mechanical loads and local and systemic biological stressors that jointly erode muscle quality. Despite shared pathways, respiratory and limb muscles diverge phenotypically: the chronically loaded diaphragm becomes more oxidative and fatigue-resistant but force-limited by geometry, while underloaded locomotor muscles drift toward atrophy, glycolytic predominance and myosteatosis. An integrated *Damage–Regeneration–Remodeling* approach is therefore recommended: *reduce damage* (e.g., lung deflation, oxygen/NIV, prudent steroid use), *enable regeneration* (e.g., combined resistance–endurance training with nutritional/endocrine support), and *steer remodeling* (e.g., sustain activity after pulmonary rehabilitation, monitor and target muscle structure and function, add inspiratory muscle training when needed). Implemented in phenotype-aware individualized programs, this strategy offers a coherent path to improve function and durability of gains across both muscle compartments.

### Future Directions

Despite major advances in understanding the cellular and molecular basis of skeletal muscle dysfunction in COPD, several critical questions remain unresolved. Future research should focus on the local actions of myokines and related paracrine mediators, which likely play a pivotal role in orchestrating the interplay between injury, inflammation and regeneration within respiratory and locomotor muscles. Clarifying these local signaling circuits could reveal new molecular targets to promote effective repair and remodeling.

Integrating emerging knowledge on mitochondrial quality control, autophagy, and endoplasmic reticulum stress into a more cohesive framework of proteostasis and energy failure is another pressing need. Whether these pathways can be therapeutically modulated (through exercise, nutritional, or pharmacological interventions, or via novel bioactive compounds with antioxidant or anabolic potential) remains an open question. In particular, the effects of growth hormone secretagogues, myostatin/activin pathway inhibitors, and biological agents already and progressively used in COPD (e.g., anti–IL-5 or anti–IL-4R therapies) on skeletal muscle structure and function deserve dedicated investigation.

Future studies should also explore the interactions between COPD-related muscle dysfunction, aging, and multimorbidity, as these processes share convergent catabolic and inflammatory pathways that may amplify systemic frailty. Equally important, sex-related differences in the mechanisms and manifestations of muscle dysfunction remain virtually unexplored and could uncover distinct biological and hormonal modulators with therapeutic relevance.

Another area requiring attention is biomass-smoke-related COPD, an increasingly prevalent phenotype in low- and middle-income countries. Whether its molecular signatures and downstream consequences on skeletal muscle differ from those of tobacco-induced disease is unknown.

Finally, advancing toward mechanism-based, phenotype-guided therapies will require integrative longitudinal studies combining multi-omics profiling, imaging, and functional assessments. Such approaches can capture the dynamic nature of the Damage–Regeneration–Remodeling cycle, identify biomarkers predictive of reversibility, and guide precision interventions aimed at preventing or reversing muscle dysfunction in COPD.

## Figures and Tables

**Table 1 cells-14-01731-t001:** Damage–Repair–Remodeling framework related to Muscle Dysfunction in COPD.

Domain	Mechanism (Examples)	Key Markers/Assays	Typical Pattern	Clinical Readouts
Damage RM	Mechanical load/Geometry (hyperinflation, PEEPi) Oxidative/nitrosative stress	MIP/SNIP, Pdi ↓ Diaphragm thickness ↓ (US) Length–tension disadvantage Protein carbonyls; 4-HNE; nitrotyrosine	Sarcolemmal damage Sarcomere damage Oxidative signatures	Dyspnea ↑; Strength ↓; Endurance ↓ Exercise limitation (6MWD and CPET ↓)
Damage LM	Oxidative/nitrosative stress Proteolysis Proteostasis failure Mitochondrial/ER stress Microvascular involvement NMJ involvement Lipotoxicity/myosteatosis	Protein carbonyls; 4-HNE; nitrotyrosine Atrogin-1, MuRF-1, LC3-II; ubiquitinated proteins; p62; calpain/caspase activity CS/SDH/COX ↓; mtDNA damage BiP/CHOP (UPR) Capillary-to-fiber ratio ↓ Changes in NMJ morphology Neurogenic grouping EMG abnormalities CT attenuation; MRI PDFF; Lipid intermediates	Oxidative signatures Predominant proteolysis Oxidative capacity ↓ Oxidative capacity ↓ NMJ instability Fat infiltration	Weakness Fatigability Muscle mass ↓ Poor muscle quality Strength ↓; Endurance ↓ Exercise limitation (6MWD and CPET ↓) Early acidosis Impaired O_2_ kinetics
Repair RM and LM	Satellite cells/myogenesis Inflammatory resolution Anabolic signaling	Pax7^+^ cells MyoD/myogenin Fusion indices M1→M2 polarization Cytokine panels IGF-1/AKT/mTOR Myostatin/activin	Both impaired in sarcopenia (diaphragm less studied) Impairs resolution in both Anabolic resistance in both Sex/Age effects	Injury-blunted Recovery after training Fibrosis-prone repair Persistent weakness Rehabilitation gains limited
Remodeling RM	Diaphragm conditioning	Shift to type I ↑ Mitochondrial content ↑ Capillarity ↑	Aerobic metabolism ↑	Fatigue resistance Strength not restored
Remodeling LM	Limb glycolytic shift Loss of muscle quality loss ECM/muscle fibrosis	Fiber atrophy Shift to type II Mitochondrial dysfunction Capillarity ↓ Intramuscular fat (CT/MRI) Collagen content Stiffness indices	Aerobic metabolism ↓ Myosteatosis ECM debris deposition	Weakness Strength ↓ Endurance ↓ Exercise limitation Reduced muscle quality Force transmission ↓ Post-exercise recovery ↓

Arrows indicate the direction of changes (increase, up; decrease, down).

## Data Availability

Data and technical procedures are available to interested researchers upon request.

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
