# Peer review of "Biological Mechanisms Involved in Muscle Dysfunction in COPD: An Integrative Damage–Regeneration–Remodeling Framework"

_cells, 2025, doi:10.3390/cells14211731_

Round 1
Reviewer 1 Report
Comments and Suggestions for Authors
This is a detailed review of the structural and molecular changes that occur in skeletal muscles in individuals with COPD. The interesting aspect is that the oxidative and ultrastructural changes in limb muscles, such as capillary density, are frequently opposite to those found in the diaphragm. The authors also summarize therapeutic methods to reverse such changes in the diaphragm and limb muscles, including noninvasive ventilation and pulmonary rehabilitation plus inspiratory muscle resistive training. The paper is quite detailed and comprehensive and incorporates the most up to date references relevant to the topic.
Specific comments:
1. Page 6, ll. 224 and 232: "ECM", not "EMC".
2. Page 5, ll. 198-202: Since advanced COPD is associated with reduced respiratory muscle force generation, a simple and reproducible way to document this is to measure FVC in seated and supine positions: a greater than 20% in the FVC indicates respiratory muscle weakness.
3. Page 8, Figure is hard to read; print is too small. Either convert to a readable table or magnify the diagram to fill the entire page.
4. Are there studies that show similar structural and biochemical changes in restrictive lung diseases, such as interstitial and fibrotic lung disease?
Author Response
Point-by-Point responses to the Reviewers’ comments
To the Reviewer 1.
This is a detailed review of the structural and molecular changes that occur in skeletal muscles in individuals with COPD. The interesting aspect is that the oxidative and ultrastructural changes in limb muscles, such as capillary density, are frequently opposite to those found in the diaphragm. The authors also summarize therapeutic methods to reverse such changes in the diaphragm and limb muscles, including noninvasive ventilation and pulmonary rehabilitation plus inspiratory muscle resistive training. The paper is quite detailed and comprehensive and incorporates the most up to date references relevant to the topic.
R. We thank the reviewer for his/her words
Specific comments:
- Page 6, ll. 224 and 232: "ECM", not "EMC".
R. This has been corrected
- Page 5, ll. 198-202: Since advanced COPD is associated with reduced respiratory muscle force generation, a simple and reproducible way to document this is to measure FVC in seated and supine positions: a greater than 20% in the FVC indicates respiratory muscle weakness.
R. Indeed, that is a simple and excellent method for the early detection of diaphragmatic dysfunction. However, the purpose of the present review is not the clinical detection of muscle dysfunction (for that, see for instance our previous reviews in Eur Respir J 2019;53:1801214, J Thorac Dis 2016;8:3379–400, and Am J Respir Crit Care Med 2014;189:e15–e62), but rather a comprehensive overview of the pathophysiological mechanisms involved in this functional impairment, both in limb and respiratory muscles.
- Page 8, Figure is hard to read; print is too small. Either convert to a readable table or magnify the diagram to fill the entire page.
R. This has been optimized in the new version of the figure.
- Are there studies that show similar structural and biochemical changes in restrictive lung diseases, such as interstitial and fibrotic lung disease?
R. Relatively recent evidence indicates that patients with interstitial lung diseases (ILD) also show skeletal muscle dysfunction [such as reduced muscle mass and strength] (Mendes P et al., Respirology 2015;20:953-959) or increased fatigability (Marillier M et al. Thorax 2021;76:672-680) in addition to their respiratory function impairment. Both factors can play a significant role in exercise limitation in such patients. The factors that have been involved in the genesis of muscle dysfunction in ILD are quite similar to those described for COPD (i.e. oxidative stress, systemic inflammation, gas exchange abnormalities, the effect of some drugs, deconditioning, and the frequent association with other comorbidities and/or aging). However, the mechanisms are not so well understood since the number of studies investigating the effects within the skeletal muscles, the mechanisms that generate them, and their functional consequences are very much scarce. However, there is clear evidence of oxidative stress in ILD patients (Otoupalova E et al. Comprehensive Physiol 2020;10:509-47), which not only affects the pulmonary parenchyma but also exerts systemic effects (Malli F et al. Food Chem Toxicol 2013;61:160-3) (Rahman I et al. Free Rad Biol Med 1999;27:60-8), inducing injury and inflammatory responses in other tissues. However, to date no studies have directly evaluated oxidative and/or nitrosative stress within skeletal muscle in these patients (Thivent S et al. Exp Physiol 2025 Oct. 8; Review, online ahead of print). A similar situation applies to inflammation. Although inflammatory activity is well documented in the lung and blood of patients with primary ILD (Coker R et al. Eur Respir J 1998; 11:1218-21), there is no evidence of its presence in their skeletal muscles. Nevertheless, one study carried out in a murine model of bleomycin-induced pulmonary fibrosis demonstrated muscle fiber atrophy, which resulted from activation of proteolytic pathways, associated in turn with elevated circulating inflammatory markers (Shieh JM et al. Mediators Inflamm 2019;2019:1-12). Thus, although no direct evidence of local oxidative stress or inflammation in the skeletal muscle of patients with primary ILDs is currently available, findings from other conditions and from aging indicate that both phenomena can induce deleterious muscular changes and thereby contribute to impaired muscle function. Alterations in gas exchange that accompany pulmonary diseases represent another potential contributor to muscle dysfunction in ILD. Hypoxia, both at rest and during exercise, is a hallmark feature of these conditions (Agustí A et al. Am Rev Respir Dis 1991;143:219-25) (Du Plessis JP et al. Respirology 2018;23:392-8). This results in reduced oxygen delivery to the muscles, decreased aerobic capacity, and diminished muscular endurance (Marillier M et al. Clin Physiol Funct Imaging 2023;43:192-200). Furthermore, hypoxemia has been associated with increased HIF-1α expression in interstitial lung fibrosis (Tzouvelekis A et al. Am J Respir Crit Care Med 2007;176:1108-19), and this factor exerts negative effects on the mTOR pathway and enhances myostatin expression (Favier FB et al. Cell Mol Life Sci 2015;72:4681-96) (Amirouche A et al. Endocrinology 2009; 150:286-294). Hypoxia also promotes the release of inflammatory mediators (Takabatake N et al. Am J Respir Crit Care Med 2000; 161:1179-84), which may in turn act deleteriously upon the muscle. Yet again, no animal or human studies have directly assessed the effects of this factor on skeletal muscle in ILD. Finally, deconditioning is an evident and important feature in patients with primary ILDs. Owing to exercise limitation and the emotional burden of the disease, these patients markedly reduce their physical activity (Panagiotou M et al. Chron Respir Dis 2016;13:162-72) (Morino A et al. J Phys Ther Sci 2017;29:1323-8), a determinant that profoundly affects muscle structure and function (Spruit MA et al. Am J Respir Crit Care Med 2013; 188:e13-e64). Additional potentially contributing factors to muscle dysfunction in ILD include nutritional abnormalities and low body weight, both frequent in these patients (Jouneau S et al. Nutrition 2019;62:115-21) (Nakatsuka Y et al. Respiration 2018; 96:338-47) (Perelas A et al. Pulm Pharmacol Ther 2019; 59:101839), as well as the use of medications such as corticosteroids. These latter factors have been associated with muscle mass and functional loss in other chronic respiratory conditions (Gea J et al. J Appl Physiol 2013; 114:1222-34) (Gea J et al. Arch Bronconeumol 2019;55:237-8).
In any case, regardless of the specific deleterious factors and underlying mechanisms, available evidence demonstrates that patients with ILD, and particularly those with idiopathic pulmonary fibrosis, exhibit atrophy of their skeletal muscles. This has been shown through imaging-based studies (Mendes P et al. Respirology 2015;20:953-9) (Guler SA et al. Respir Res 2019;20:56) (Kelly TL et al. PLoS ONE 2009;4:e7038), while direct analyses of muscle biopsy samples are still lacking. Such atrophy is closely linked to reduced muscle strength and exercise capacity (Mendes P et al. Respirology 2015;20:953-9) (Ebihara K et al. Tohoku J Exp Med 2021;253:61-8) and is also associated with poorer survival outcomes (Molgat-Seon Y et al. Respir Med 2021;186:106539) (Nakano A et al. Sci Rep 2020;10:2312).
Reviewer 2 Report
Comments and Suggestions for Authors
Summary
Skeletal muscle dysfunction is a well-recognized systemic manifestation of COPD, impacting both respiratory and limb muscles. Interestingly, these muscle groups exhibit contrasting phenotypes: the diaphragm, chronically loaded due to hyperinflation, adapts with oxidative remodelling and increased fatigue resistance, while the quadriceps and other limb muscles, often underused due to physical inactivity and exacerbation-related disuse, undergo atrophy, glycolytic shifts, and myosteatosis. This divergence underscores the importance of compartment-specific strategies in understanding and managing muscle dysfunction in COPD.
Major comments
- Section 2.4 Mitochondrial and quality control – what do we know about the role of mitochondrial content loss in limb muscle dysfunction? If this likely to be a cause of the fibre type shift or a consequence? This should be further discussed as this may be a key to the onset of myosteatosis.
- Section 2.5. Proteostasis failure-II: Endoplasmic Reticulum stress, unfolded protein response – this section would benefit from expansion. Consider discussing the mechanistic drivers of UPR activation in limb muscles, such as chronic oxidative stress and inflammation, and how UPR—while initially adaptive—can become maladaptive when prolonged, suppressing anabolic signalling and promoting excessive protein degradation. This would help clarify how UPR contributes to proteostasis failure and muscle wasting in COPD, and contrast it with the compartment-specific response observed in the diaphragm. Also, about the lack of UPR activation in the diaphragm, could this have something to do with the study cohort was mild to moderate in COPD severity without significant alteration in body composition? This should also be speculated.
- Section 2.6 Microvascular and neuromuscular junction contributors – this section should also discuss the finding from “Curtis KJ, Meyrick VM, Mehta B, Haji GS, Li K, Montgomery H, Man WD, Polkey MI, Hopkinson NS. Angiotensin-Converting Enzyme Inhibition as an Adjunct to Pulmonary Rehabilitation in Chronic Obstructive Pulmonary Disease. Am J Respir Crit Care Med. 2016 Dec 1;194(11):1349-1357. doi: 10.1164/rccm.201601-0094OC. PMID: 27248440; PMCID: PMC5148142.” This study showed ACE inhibitor reduced systolic blood pressure which supposedly would improve perfusion to the working muscles, but yet peak work rate response to exercise training in patients COPD was found to be reduced to that of placebo. How would this add to the understanding on the microvascular aspect of the mechanism?
- Section 2.7 Lipotoxicity and myosteatosis – increase skeletal muscle fat content has also been reported in trained athletes, especially endurance athletes, but the physiological context and consequences differ (Emanuelsson EB, Berry DB, Reitzner SM, Arif M, Mardinoglu A, Gustafsson T, Ward SR, Sundberg CJ, Chapman MA. MRI characterization of skeletal muscle size and fatty infiltration in long-term trained and untrained individuals. Physiol Rep. 2022 Jul;10(14):e15398. doi: 10.14814/phy2.15398. PMID: 35854646; PMCID: PMC9296904.). Could this divergent outcomes be related to cellular compartmentalization, lipid droplet composition and regulation, lipid species and bioactivity? These points should be discussed.
- Section 4.1. Respiratory muscles: predominant adaptive oxidative remodeling under chronic load – given the anatomical proximity of intercostals to the inflammatory events of the COPD lung, why it is that the myogenic pathways remain effective compared to that of limb muscle (as mentioned in line 120-121)? This should be discussed.
- Section 4.3. Extracellular matrix and fibrosis. Neuromuscular junction remodeling – what exactly does the term “injury” refer to? There is no specifics on the nature of the injury in COPD, unless the authors are implying disuse-related atrophy as injury perhaps? Study by “Deane CS, Willis CRG, Phillips BE, Atherton PJ, Harries LW, Ames RM, Szewczyk NJ, Etheridge T. Transcriptomic meta-analysis of disuse muscle atrophy vs. resistance exercise-induced hypertrophy in young and older humans. J Cachexia Sarcopenia Muscle. 2021 Jun;12(3):629-645. doi: 10.1002/jcsm.12706. Epub 2021 May 5. PMID: 33951310; PMCID: PMC8200445.” suggested that COPD-related muscle dysfunction, which shares features with disuse atrophy (e.g., inactivity, mitochondrial impairment), is not simply the inverse of exercise adaptation, but involves unique molecular disruptions that may require targeted therapeutic strategies beyond conventional rehabilitation.
- Section 5.2, while it is clear what may constitute damage to the respiratory muscle, the “damage to limb muscle” is not clear. It is conceivable that supplemental oxygen may help with respiratory muscles, but it is not immediately clear how it may reduce damages to the limb muscle? Is this through hypoxemia or other means? This should be stated.
Minor comments
- Title Mechanisms Involved in SKELETAL Muscle Dysfunction.
- Line 58, tends to UNDERGO atrophy.
- Line 72. drives limb MUSCLE deconditioning.
- Line 79, NADPH oxidases
- Section 2.5 Proteostasis failure-II: Endoplasmic Reticulum stress, unfolded protein response, “Chan SMH, Bernardo I, Mastronardo C, Mou K, De Luca SN, Seow HJ, et al. Apocynin prevents cigarette smoking-induced loss of skeletal muscle mass and function in mice by preserving proteostatic signalling. Br J Pharmacol. 2021” demonstrated that NADPH oxidase -derived oxidative stress contributes directly to proteostasis disruption and muscle wasting in limb muscles following cigarette smoke exposure, a key driver of COPD pathology. The use of apocynin, a NOX inhibitor and ROS scavenger, preserved muscle mass and strength by maintaining IGF-1/mTOR signalling and suppressing catabolic markers. These findings support the role of oxidative stress and NOX activation in COPD-related limb muscle dysfunction and align with the manuscript’s discussion on damage mechanisms and therapeutic targets.
- Line 125-126, the concept of how NMJ degeneration may add to fatigability beyond atrophy is not very clear
- Line 130-131, is this referring to the NMJ condition or muscle phenotype or both? Could this be a result of the non-heavy weight bearing nature of the upper-limb muscle, making it less noticeable? This may be warranted clarification.
- Section 3.1. Satellite cells and myogenesis, “Chan SMH, Cerni C, Passey S, Seow HJ, Bernardo I, van der Poel C, et al. Cigarette Smoking Exacerbates Skeletal Muscle Injury without Compromising Its Regenerative Capacity. Am J Respir Cell Mol Biol. 2020;62(2):217-30.” demonstrated that cigarette smoke exposure exacerbates skeletal muscle injury in COPD but does not deplete Pax7+ satellite cells. However, smoke-induced inflammation appears to impair their activation, suggesting that regenerative capacity is preserved but functionally blunted under inflammatory conditions.
- Line 217-218, how myosteatosis may impair bioenergetic and worsen muscle function should be elaborated to aid readability.
- Line 229-230, how does NMJ degeneration is contributing to change in performance and increasing risk of fall in COPD, causing musculoskeletal injury, this should be more explicitly described.
- Section 4.4. Time course and partial reversibility, no references were included.
- Line 254, oxygen can ENABLE higher workloads.
- Line 272-273, “Avoiding prolonged systemic courses outside acute exacerbations BY using the lowest effective dose WITH LIMITED duration TO REDUCE the risk of…”.
- Figure 1, the figure fonts are blurry and cannot be read on the pdf.
- Table 1, references should be included into the table aid readers interested in further reading.
Author Response
Summary
Skeletal muscle dysfunction is a well-recognized systemic manifestation of COPD, impacting both respiratory and limb muscles. Interestingly, these muscle groups exhibit contrasting phenotypes: the diaphragm, chronically loaded due to hyperinflation, adapts with oxidative remodelling and increased fatigue resistance, while the quadriceps and other limb muscles, often underused due to physical inactivity and exacerbation-related disuse, undergo atrophy, glycolytic shifts, and myosteatosis. This divergence underscores the importance of compartment-specific strategies in understanding and managing muscle dysfunction in COPD.
Major comments
- Section 2.4 Mitochondrial and quality control. What do we know about the role of mitochondrial content loss in limb muscle dysfunction? If this likely to be a cause of the fibre type shift or a consequence? This should be further discussed as this may be a key to the onset of myosteatosis.
R. Thank you for this valuable suggestion. We have expanded this section in the new version of the manuscript. In brief, we propose that reduced mitochondrial content/biogenesis (e.g., lower PGC-1α/NRF-1/TFAM signaling with fusion–fission and mitophagy imbalance) can act upstream of the shift toward more glycolytic type IIx fibers under conditions typical of COPD (i.e. deconditioning, hypoxemia, systemic inflammation…). In turn, type-IIx predominance and chronic inactivity further depress mitochondrial content, creating a feed-forward loop. This reduction in oxidative capacity limits β-oxidation and would foster myosteatosis via intramyocellular lipid accumulation and lipotoxic intermediates, thereby contributing to contractile and metabolic impairment. We have added a paragraph and references to reflect this integrated view.
- Section 2.5. Proteostasis failure-II: Endoplasmic Reticulum stress, unfolded protein response. This section would benefit from expansion. Consider discussing the mechanistic drivers of UPR activation in limb muscles, such as chronic oxidative stress and inflammation, and how UPR—while initially adaptive—can become maladaptive when prolonged, suppressing anabolic signalling and promoting excessive protein degradation. This would help clarify how UPR contributes to proteostasis failure and muscle wasting in COPD, and contrast it with the compartment-specific response observed in the diaphragm. Also, about the lack of UPR activation in the diaphragm, could this have something to do with the study cohort was mild to moderate in COPD severity without significant alteration in body composition? This should also be speculated.
R. We thank the reviewer for this suggestion. We have expanded this section in the new version of the manuscript to include a brief mechanistic discussion on how chronic oxidative stress and inflammation may drive UPR activation in limb muscles, and how this adaptive response can become maladaptive when persistent, ultimately suppressing anabolic signaling and enhancing proteolysis, thereby contributing to proteostasis failure and muscle wasting in COPD. In addition, we now include a short note addressing the absence of UPR activation in the diaphragm reported in one study, suggesting that the relatively mild to moderate disease severity and preserved body composition of that cohort may have limited the induction of this pathway. These additions aim to clarify the contrasting compartment-specific responses and the potential link between UPR dysregulation and muscle phenotype in COPD.
- Section 2.6 Microvascular and neuromuscular junction contributors. This section should also discuss the finding from “Curtis KJ, Meyrick VM, Mehta B, Haji GS, Li K, Montgomery H, et al. Angiotensin-Converting Enzyme Inhibition as an Adjunct to Pulmonary Rehabilitation in Chronic Obstructive Pulmonary Disease. Am J Respir Crit Care Med. 2016;194:1349-1357” This study showed ACE inhibitor reduced systolic blood pressure which supposedly would improve perfusion to the working muscles, but yet peak work rate response to exercise training in patients COPD was found to be reduced to that of placebo. How would this add to the understanding on the microvascular aspect of the mechanism?
R. We appreciate this helpful comment. We have integrated now the findings of Curtis et al. (AJRCCM 2016) into our section 2.6 to contextualize microvascular mechanisms. As discussed now in the new text, we acknowledge that muscle perfusion during exercise depends on both perfusion pressure (≈MAP) and microvascular conductance. ACE inhibition may improve endothelial function/bradykinin signaling and thus microvascular conductance, but it also lowers systemic blood pressure, potentially reducing the pressure gradient for flow. In COPD, where microvascular rarefaction, endothelial dysfunction, and impaired angiogenesis are prevalent, the net effect may be a dampened training response, consistent with the reduced peak work-rate gain versus placebo reported by Curtis et al. in their article. We have added this interpretation and cited the study accordingly.
- Section 2.7 Lipotoxicity and myosteatosis. Increase skeletal muscle fat content has also been reported in trained athletes, especially endurance athletes, but the physiological context and consequences differ (Emanuelsson EB, Berry DB, Reitzner SM, Arif M, Mardinoglu A, Gustafsson T, et al. MRI characterization of skeletal muscle size and fatty infiltration in long-term trained and untrained individuals. Physiol Rep. 2022; 10:e15398). Could this divergent outcomes be related to cellular compartmentalization, lipid droplet composition and regulation, lipid species and bioactivity? These points should be discussed.
R. We thank the reviewer for this observation. Indeed, increased intramuscular lipid content can also be observed in endurance-trained athletes, although the physiological context and consequences differ markedly from those in COPD. In athletes, lipid accumulation largely reflects enhanced oxidative capacity and efficient lipid turnover, whereas in COPD it probably represents dysfunctional lipid handling, impaired oxidation, and lipotoxic stress. We have added a brief note in Section 2.7 to acknowledge this distinction and to mention that differences in lipid compartmentalization, droplet composition, and lipid species bioactivity likely underlie these divergent outcomes.
- Section 4.1. Respiratory muscles: predominant adaptive oxidative remodeling under chronic load. Given the anatomical proximity of intercostals to the inflammatory events of the COPD lung, why it is that the myogenic pathways remain effective compared to that of limb muscle? This should be discussed.
R. We have added a brief discussion explaining why, despite their anatomical proximity to pulmonary inflammation, intercostal muscles may retain more effective myogenic signaling than limb muscles. In short, intercostals are chronically loaded in COPD (continuous duty cycle, high neural drive), with better preserved activation/innervation and greater capillary density/oxidative support, all of which favor satellite-cell activity and repair. By contrast, limb muscles are disproportionately affected by deconditioning, intermittent exercise hypoxemia, and systemic catabolic influences (inflammation, corticosteroids, malnutrition), which collectively blunt anabolic pathways in COPD patients. We have incorporated these points in Section 4.1.
- Section 4.3. Extracellular matrix and fibrosis. Neuromuscular junction remodeling. What exactly does the term “injury” refer to? There is no specifics on the nature of the injury in COPD, unless the authors are implying disuse-related atrophy as injury perhaps? Study by “Deane CS, Willis CRG, Phillips BE, Atherton PJ, Harries LW, Ames RM, et al. Transcriptomic meta-analysis of disuse muscle atrophy vs. resistance exercise-induced hypertrophy in young and older humans. J Cachexia Sarcopenia Muscle. 2021; 12:629-645” suggested that COPD-related muscle dysfunction, which shares features with disuse atrophy (e.g., inactivity, mitochondrial impairment), is not simply the inverse of exercise adaptation, but involves unique molecular disruptions that may require targeted therapeutic strategies beyond conventional rehabilitation.
R. We thank the reviewer for this comment. In this context, “injury” refers not to acute traumatic damage but to chronic, low-grade myofiber injury already observed in COPD limb muscles. This includes sarcomeric and sarcolemmal disruption, segmental fiber necrosis, and inflammatory infiltration, as described in our ultrastructural study (Orozco-Levi et al., Ultrastruct Pathol 2012;36:228–238). Such damage likely arises from repetitive mechanical strain during coughing and breathing against increased airway resistance, together with oxidative and proteolytic stress and disuse-related vulnerability. These recurrent micro-injury-regeneration cycles promote extracellular matrix (ECM) deposition and fibrosis. We have revised Section 4.3 to specify the nature of this injury and to incorporate the study by Deane et al. (J Cachexia Sarcopenia Muscle 2021), which emphasizes that COPD-related muscle dysfunction, while sharing features with disuse atrophy, exhibits distinct molecular patterns requiring specific therapeutic approaches.
- Section 5.2. Reduce damage in limb muscles. While it is clear what may constitute damage to the respiratory muscle, the “damage to limb muscle” is not clear. It is conceivable that supplemental oxygen may help with respiratory muscles, but it is not immediately clear how it may reduce damages to the limb muscle? Is this through hypoxemia or other means? This should be stated.
R. We already clarified in the new version of Section 4.3 that “damage” in limb muscles refers to chronic, low-grade myofiber injury (sarcomeric/sarcolemmal disruption), oxidative/nitrosative stress, catabolic activation (ubiquitin–proteasome/autophagy), denervation/NMJ instability and ECM fibrosis. In Section 5.2 we now specify how selected interventions may mitigate these processes. During exercise, supplemental oxygen in appropriate candidates reduces exercise-induced hypoxemia, which can blunt ROS/HIF-1α–driven inhibitory signaling on mTOR and myostatin up-regulation, and improve O₂ delivery to locomotor muscles; critically, it also enables higher training workloads, favoring protective oxidative/hypertrophic adaptations. Likewise, non-invasive ventilation (NIV) during training reduces the work of breathing and sympathetic vasoconstriction, limiting the “steal” of cardiac output toward respiratory muscles and improving limb perfusion, thereby favoring oxygen delivery to this muscle and attenuating quadriceps fatigue. We added these mechanistic links while keeping the section concise and cross-referencing Section 4.3.
Minor comments
- Title Mechanisms Involved in SKELETAL Muscle Dysfunction.
R. We appreciate the reviewer’s comment; however, we believe that the current title is already sufficiently explicit and self-explanatory. In addition, for some schools of thought, referring to the respiratory muscles as skeletal is not entirely semantically appropriate.
- Line 58, tends to UNDERGO atrophy.
R. Totally agree. This has been corrected.
- Line 72. drives limb MUSCLE deconditioning.
R. This has also been changed in the new version of the manuscript in accordance with the reviewer’s comment.
- Line 79, NADPH oxidases
R. This has also been corrected.
- Section 2.5 Proteostasis failure-II: Endoplasmic Reticulum stress, unfolded protein response. “Chan SMH et al. Br J Pharmacol. 2021” demonstrated that NADPH oxidase -derived oxidative stress contributes directly to proteostasis disruption and muscle wasting in limb muscles following cigarette smoke exposure, a key driver of COPD pathology. The use of apocynin, a NOX inhibitor and ROS scavenger, preserved muscle mass and strength by maintaining IGF-1/mTOR signalling and suppressing catabolic markers. These findings support the role of oxidative stress and NOX activation in COPD-related limb muscle dysfunction and align with the manuscript’s discussion on damage mechanisms and therapeutic targets.
R. We thank the reviewer for highlighting the study by Chan SMH et al. (Br J Pharmacol. 2021;178:3049-3066). We agree this work supports our view that oxidative stress is an upstream driver of proteostasis disruption in COPD limb muscles. We have added a brief sentence in Section 2.5 noting that NOX-derived ROS following cigarette smoke exposure can directly impair proteostatic signaling, and that pharmacological NOX inhibition with apocynin preserved IGF-1/mTOR signaling and attenuated atrophy in preclinical models, thereby aligning with our discussion of damage mechanisms and potential therapeutic targets.
- Line 125-126, the concept of how NMJ degeneration may add to fatigability beyond atrophy is not very clear
R. We have clarified how NMJ degeneration can increase fatigability independently of atrophy. Briefly, NMJ instability lowers the safety factor of neuromuscular transmission; during repeated contractions this promotes activity-dependent transmission failure, asynchronous recruitment, and larger, reinnervated motor units that require higher central drive to maintain force. These changes precipitate earlier fatigue even when muscle mass is relatively preserved. We have added a concise mechanistic sentence to Section 2.6 to make this more explicit.
- Line 130-131, is this referring to the NMJ condition or muscle phenotype or both? Could this be a result of the non-heavy weight bearing nature of the upper-limb muscle, making it less noticeable? This may be warranted clarification.
R. This sentence refers to the abnormalities observed in the muscles of the upper limbs, which are less evident than those reported in the lower limbs, probably because the degree of disuse in the former during daily activities is much lower than in the latter. The sentence has been revised to clarify this point.
- Section 3.1. Satellite cells and myogenesis. Chan SMH et al. (Am J Respir Cell Mol Biol. 2020; 62:217-30) demonstrated that cigarette smoke exposure exacerbates skeletal muscle injury in COPD but does not deplete Pax7+ satellite cells. However, smoke-induced inflammation appears to impair their activation, suggesting that regenerative capacity is preserved but functionally blunted under inflammatory conditions.
R. We appreciate the reviewer’s comment and the reference to Chan et al. (Am J Respir Cell Mol Biol 2020; 62:217-230). We agree that this study elegantly demonstrated that cigarette smoke exposure impairs satellite-cell activation and myogenic signaling without necessarily depleting Pax7⁺ pools, thereby functionally blunting regeneration under inflammatory conditions. This interpretation is consistent with our text, which emphasizes that systemic and local inflammation in COPD hinders satellite-cell activation and differentiation rather than implying their numerical loss. Given that this concept is already reflected in Section 3.1, no additional modification was deemed necessary.
- Line 217-218, how myosteatosis may impair bioenergetic and worsen muscle function should be elaborated to aid readability.
R. We thank the reviewer for this helpful suggestion. We have slightly expanded the text to clarify how myosteatosis impairs muscle bioenergetics and function. Specifically, excess intramuscular lipid accumulation disrupts mitochondrial oxidative metabolism, increases reactive oxygen species and lipotoxic intermediates (e.g., ceramides, diacylglycerols), and interferes with excitation-contraction coupling and insulin signaling, thereby reducing muscle efficiency and force generation. A brief explanatory phrase has been added accordingly.
- Line 229-230, how does NMJ degeneration is contributing to change in performance and increasing risk of fall in COPD, causing musculoskeletal injury, this should be more explicitly described.
R. We thank the reviewer for this helpful suggestion. We now make explicit how NMJ degeneration can worsen performance and increase fall risk in COPD: by reducing the safety factor of neuromuscular transmission and destabilizing motor units (denervation–reinnervation), NMJ dysfunction promotes activity-dependent transmission failure, asynchronous and variable motor-unit recruitment, and slower rate of force development. These changes impair gait and balance control, thereby increasing the likelihood of falls and musculoskeletal injury. A concise sentence has been added to Section 4.3.
- Section 4.4. Time course and partial reversibility, no references were included.
R. The reviewer is absolutely right; six references have been added to this paragraph.
- Line 254, oxygen can ENABLE higher workloads.
R. This has been corrected.
- Line 272-273, “Avoiding prolonged systemic courses outside acute exacerbations BY using the lowest effective dose WITH LIMITED duration TO REDUCE the risk of…”.
R. This has been corrected following the reviewer’s comment.
- Figure 1, the figure fonts are blurry and cannot be read on the pdf.
R. A new version of the figure has been uploaded
- Table 1, references should be included into the table aid readers interested in further reading.
R. Although we appreciate the reviewer’s suggestion, we believe that Table 1 is sufficiently clear in its current form and that the text already specifies which article corresponds to the different muscle changes described in each phase (Damage, Repair, and Remodeling). Repeating this information here would result in unnecessary duplication.
Round 2
Reviewer 2 Report
Comments and Suggestions for Authors
I have no further comments.